# Rare A360T Mutation Alters GSK3β(Ser9) Binding in the Cytosolic Loop of Presenilin 1, Influencing β-Catenin Nuclear Localization and Pro-Death Gene Expression in Alzheimer’s Disease Case

**DOI:** 10.3390/ijms242316999

**Published:** 2023-11-30

**Authors:** Michalina Wężyk, Mariusz Berdyński, Adam Figarski, Magdalena Skrzypczak, Krzysztof Ginalski, Marzena Zboch, Izabela Winkel, Cezary Żekanowski

**Affiliations:** 1Department of Neurogenetics and Functional Genomics, Mossakowski Medical Research Institute, Polish Academy of Sciences, 5 Pawińskiego Street, 02-106 Warsaw, Polandc.zekanowski@imdik.pan.pl (C.Ż.);; 2Department of Experimental Hematology, Institute of Hematology and Transfusion Medicine, 14 Indiry Gandhi Street, 02-776 Warsaw, Poland; 3Laboratory of Bioinformatics and Systems Biology, Centre of New Technologies, University of Warsaw, 93 Żwirki i Wigury Street, 02-089 Warsaw, Poland; 4Research and Education Center for Dementia Diseases in Ścinawa, Alzheimer’s Center, 12 Jana Pawła II Street, 59-330 Ścinawa, Poland; 5Faculty of Physical Education, Gdansk University of Physical Education and Sport, Górskiego 1 Street, 80-336 Gdansk, Poland

**Keywords:** Alzheimer’s disease, β-catenin/GSK3β, PS1 cytosolic loop mutation, non-amyloidogenic pathology

## Abstract

Presenilin 1 (PS1) forms, via its large cytosolic loop, a trimeric complex with N-cadherin and β-catenin, which is a key component of Wnt signaling. PS1 undergoes phosphorylation at 353 and 357 serines upon enhanced activity and elevated levels of the GSK3β isoform. PS1 mutations surrounding these serines may alter the stability of the β-catenin complex. Such mutations are found in some cases of familial early-onset Alzheimer’s disease (fEOAD), but their functional impact remains obscure. One of such variants of PS1, the A360T substitution, is located close to GSK3β-targeted serine residues. This variant was recently demonstrated in the French population, but more detail is needed to understand its biological effects. To assess the significance of this variant, we employed functional studies using a fibroblast cell line from an Alzheimer’s disease case (a female proband) carrying the A360T mutation. Based on functional transcriptomic, cellular, and biochemical assays, we demonstrated atypically impaired β-catenin/GSK3β signaling in the A360T patient’s fibroblasts. In detail, this was characterized by a decreased level of active cytosolic β-catenin and bound by PS1, an increased level of nuclear β-catenin, an increased level of inhibited GSK3β phosphorylated on Ser9, and enhanced interaction of GSK3β(Ser9) with PS1. Based on the transcriptomic profile of the A360T fibroblasts, we proposed a dysregulated transcriptional activity of β-catenin, exemplified by increased expression of various cyclin-dependent kinases and cyclins, such as cyclin D1, potentially inducing neurons’ cell cycle re-entry followed by apoptosis. The A360T cells did not exhibit significant amyloid pathology. Therefore, cell death in this PS1 cytosolic loop mutation may be attributed to impaired β-catenin/GSK3β signaling rather than amyloid deposition per se. We further estimated the biological and clinical relevance of the A360T variant by whole exome sequencing (WES). WES was performed on DNA from the blood of an A360T female proband, as well as an unrelated male patient carrying the A360T mutation and his mutation-free daughter (both unavailable for the derivation of the fibroblast cell lines). WES confirmed the highest-priority AD causality of the A360T variant in PS1 and also profiled the pathways and processes involved in the A360T case, highlighting the greatest importance of altered Wnt signaling.

## 1. Introduction

Mutations in presenilin 1 gene (*PSEN1*) are a common cause of familial early-onset Alzheimer’s disease (fEOAD). The pathogenic mechanism likely involves an enhanced proteolytic cleavage of the amyloid precursor protein (APP), generating a longer form of the β-amyloid peptide, Aβ1-42 [1]. Presenilin 1 (PS1) is a ubiquitously expressed multipass transmembrane protein. Although mainly localized to the endoplasmic reticulum and Golgi membranes, it has also been detected in the nuclear envelope and plasma membrane in complexes with catenin, which has been shown to modulate the Wnt/β-catenin signalling pathway critical for neuronal commitment and survival in neurodegenerative diseases [2].

The Wnt protein is inactive in the “off” state of the canonical Wnt/β-catenin signaling pathway, and the absence of Wnt signaling allows the assembly of glycogen synthase kinase 3 (GSK3) with axin and the adenomatous polyposis coli protein (APC) (axin/APC/GSK3 complex). As a result, active GSK3 phosphorylates cytosolic β-catenin and targets it for ubiquitination and degradation. The “off” state shuts down β-catenin-driven gene expression. In the “on” state, Wnt initiates a signaling cascade that leads to the dissociation of the axin/APC/GSK3 complex from β-catenin, which further allows translocation of β-catenin to the nucleus and induces downstream gene expression [3]. GSK3 is constitutively active under basal conditions and is inhibited by Akt-mediated phosphorylation of its N-terminal portion on serine 9 in the GSK3β isoform and serine 21 in GSK3α. Overall, the presence of inhibited forms of GSK3 (i.e., GSK3β(Ser9) or GSK3α(Ser21)) should result in allowing β-catenin downstream gene expression. Although the functions of the two isoforms largely overlap, GSK3β interacts preferentially with PS1 and is more closely associated with the pathology of Alzheimer’s disease [4,5]. GSK3β phosphorylated at serine 9 is unable to bind β-catenin, which is then not degraded and accumulates in the cytosol, which results in the continuation of its nuclear activity. Then, unphosphorylated cytosolic β-catenin translocates to the nucleus, where it participates in Wnt-responsive transcription of target genes involved in apoptosis and cell cycle regulation, including those encoding c-myc and cyclin D1. Phosphorylated β-catenin remains in the cytosol and is inactive in inducing gene expression. The constitutively (over)active GSK3β/β-catenin/Wnt signalling in the “on” state is currently recognized to promote AD [6]. In accordance with the presented work, we demonstrate the predominance of the active non-phosphorylated form of GSK3β in the fEOAD cases studied. In this report, we focused on the significance of the substitution in the cytosolic loop of PS1 at A360T, detected for the first time by French researchers [7]. In this paper, we propose alternative pathomechanisms of AD atypical mutations in PS1 involving Wnt, GSK3β, and β-catenin signaling. We found that in cultured fibroblasts from A360T patients, GSK3β was predominantly present in its inhibited phosphorylated state, which paradoxically was accompanied by a depletion of the active pool of β-catenin.

The entire concept described above was based on direct biochemical studies conducted on skin fibroblast lines derived from a patient with the aforementioned A360T mutation. Nevertheless, the effects of mutations and variants located in the cytosolic loop of PS1 are still very unclear. Therefore, we also supported our research with RNA sequencing and whole exome sequencing (WES) to pinpoint the various pathways and processes altered by the variants. 

## 2. Case Presentation

### 2.1. Materials and Methods

Ethical issues. The Ethics Committee of the Department of Neurology of the Central Clinical Hospital of the Ministry of Interior Affairs and Administration in Warsaw has approved the protocol for the acquisition of skin biopsies (decision no. 31/2013). Written informed consent for participation in this study and publication has been obtained from the patients or their legal representatives and controls, according to the Helsinki Declaration (BMJ 1991; 302:1194). 

### 2.2. Patients

A female patient with the age of onset of 55 years manifested typical early-onset Alzheimer’s disease clinical symptoms, including extrapyramidal signs, behavioral and psychiatric symptoms (agitation, depression, delusion, and hallucinations), aphasia, and cerebellar signs. At the final stage of the disease, the patient showed mutism and remained lying down and completely inactive. A skin biopsy was taken at the age of 60. The patient died at the age of 63. The DNA patient ID was 5241. This female proband was a source for fibroblasts and blood samples for RNAseq and WES analyses, respectively. MRI scan of the head of the patient taken at the beginning of the disease at 57 years old presented moderate atrophy of the brain tissue (Figure 1E).

A male patient has had slowly increasing memory disorders since 2008 (at the age of 60), which were joined by behavioral disorders and features of the extrapyramidal syndrome (psychomotor retardation, gait disturbance, hypomimia, micrograph, propulsion, plastic muscle tone in the left upper limb). The MMSE of the patient was 27/30, and the clinical diagnosis was mixed dementia in AD. The DNA patient ID was 7677. This male patient was a source of blood samples for WES analyses. For the WES study, a mutation-free daughter of the 7677 patient was also used, and the DNA ID was 7676. The 7676 patient complained of memory problems that had been getting worse over the last year (estimated by 2021), but which do not impede everyday functioning. Assessment based on the MMSE with a score of 27/30 points was considered a healthy norm, and the neurological examination showed no abnormalities. Possible autosuggestion caused by the father’s illness. MRI scan of the head of the patient taken at the beginning of the disease at 62 years old presented moderate atrophy of the brain tissue (Figure 1F).

Control subjects. Healthy subjects were screened for (1) genetic status, confirming no *PSEN1*, *PSEN2*, and *APP* mutations; (2) cognitive condition, confirming a normal score of the Mini-Mental State Examination (MMSE); (3) familial history, confirming no dementia or related disorders present in the family for at least three generations; and (4) excluded for strokes and other neurological diseases [8].

Mutation detection. The mutation was detected by Sanger sequencing of genomic DNA extracted from peripheral blood leukocytes using a standard salting-out method, as described elsewhere [8].

Cell culture. Primary fibroblasts were cultured as before [8]. 

RNA isolation, cDNA library preparation, and sequencing. Total RNA was isolated from primary fibroblast cell lines of patients and controls using the RNeasy Mini Kit (Qiagen) according to the manufacturer’s protocol. RNA quantity and purity were determined, respectively, using Qubit 2.0 using the RNA BR Assay Kit and a Bioanalyzer 2100 (Agilent) using the RNA 6000 Pico Kit, and only samples with the RNA integrity number (RIN) > 8 were taken for further analyses. Total RNA (500 ng) was converted to cDNA libraries using TruSeq Stranded Total RNA with Ribo-Zero kit (Illumina, Analityk Genetyka Sp.k., Warsaw, Poland) according to the manufacturer’s protocol. Libraries were assessed qualitatively on a Bioanalyzer 2100 using a High Sensitivity DNA Kit and quantitatively on a CFX96 Real-Time PCR system (Bio-Rad) using the KAPA Library Quantification Kit (Kapa Biosystems), and then sequenced at 2 × 76 bp on a HiSeq2500 Illumina platform. At least 20 million reads per sample were obtained with a mean quality score (Q30) > 94%. The sequencing data were demultiplexed, converted to FASTQ files, and deposited at the Sequence Read Archive NCBI NIH database as BioProject PRJNA382346.

Whole exome sequencing. DNA samples were extracted and libraries were constructed using the Twist Library Preparation EF Kit 2.0 (EUK) (Twist Bioscience, cat. no. 104381) and Twist Universal Adapter System–TruSeq Compatible, 16 Samples (Twist Bioscience, cat. no. 101307). Fragmentation conditions were applied for 20 min at 37 °C and enriched by seven cycles of amplification. The resulting libraries were subjected to fragment length control using an Agilent TapeStation 2200 analyzer and High Sensitivity D1000 ScreenTape (Agilent, cat. no. 5067-5584) and High Sensitivity D1000 Reagents (Agilent, cat. no. 5067-5585). The libraries were then enriched using the Equinox Library Amp Mix (Twist Bioscience, cat. no. 104107) and Twist Exome 2.0 plus Comp Exome spike-in (EUK) (Twist Bioscience, cat. no. 105228) reagents. The fragment lengths of the obtained library pool were determined using an Agilent TapeStation 2200 analyzer and the same reagents. The manufacturer’s protocols were followed. Sequencing was performed on an Illumina No-vaSeq 6000 instrument using NovaSeq 6000 S1 Reagent Kit v 1.5 (200 cycles) (Illumina, cat. no. 20028318), in a pair-end 2 × 100 cycle mode with the standard procedure recommended by the manufacturer and 1% addition of a Phiχ control library (Illumina, cat. no. FC-110-3001). High-quality sequencing data were obtained, and more than 94% of the data exceeded Phred Score Q37 in the amount of 119 million reads on average. Quality reports are presented in the multiqc_report.html file in Appendix A.

Bioinformatics. The RNAseq data were analyzed following the protocols described before [8]. Briefly, genes differentially expressed between patients and controls were identified using the Deseq2 package in the R Bioconductor Environment (Love, 2010). Transcripts found to be differentially expressed (fold change  ≥  2, FDR  ≤  5%, *p*-value  ≤  0.01) were summarized in heatmaps, volcano plots, MA plots, and dispersion plots, and were subjected to principal component analysis (PCA). Differentially expressed genes were analyzed functionally using the Ingenuity Pathway Analysis (IPA) software version 42012434 (http://www.ingenuity.com (accessed on 27 September 2018)) and Reactome tools (http://www.reactome.org (accessed on 22 September 2023)). Significant canonical pathways were filtered according to the IPA algorithms at a −log *p*-value cutoff  =  1.3 in a right-tailed Fisher’s exact test score. For variant calling from WES, the variant-calling pipeline from gatk4 was applied (https://gencore.bio.nyu.edu/variant-calling-pipeline-gatk4/ (accessed on 22 September 2023)). Mapping and variant calling were performed against the GRCh38.p13 human genome assembly (Ensembl). Advanced variant filtering was performed using VariantRecalibrator, ApplyVQSR, and hard filtering (https://gatk.broadinstitute.org/hc/en-us/articles/360035531112--How-to-Filter-variants-either-with-VQSR-or-by-hard-filtering#2 (accessed on 22 September 2023)) with additional criteria of sequencing coverage DP ≥ 10 and with Bcftools filtering at the minor allele frequency (MAF) of 0.02. The prioritization of variants was conducted using the Exomiser software suite version 13.1.0. (https://exomiser.github.io/Exomiser/manual/7/exomiser/ (accessed on 22 September 2023)) with or without pre-defining Human Phenotype Ontology (HPO) terms related to neurodegeneration (Appendix A), according to the attached script files (SM File S1. *A360T_HPO.yml* or SM File S2. *A360T_noHPO.yml*).

Protein cell extracts and immunoblotting. To obtain total cell lysates, fibroblasts were cultured to 100% confluence and, depending on the experiment, treated or not with 2 μM doxorubicin (Santa-Cruz Biotechnology, sc-200923) for 6, 16, 24, or 30 h. Cells were collected by trypsinization and centrifugation at 300× *g* for 5 min. After washing twice with PBS, cell pellets were lysed with standard RIPA (Radio Immuno Precipitation Assay) buffer supplemented with protease inhibitor cocktail (Roth), phosphatase inhibitor cocktail II (Sigma-Aldrich), and 20 mM NaF and sonicated for 10 cycles of 0.5 s each at 60% power (Bandelin Sonoplus HD 2070). For immunoblotting, cell lysates were boiled for 5 min at 95 °C in Laemmli buffer, separated by SDS–PAGE on 6–12% acrylamide gradient gels, and transferred to a nitrocellulose membrane (Bio-Rad). The membrane was then blocked in 5% non-fat dry milk in TBST (100 mM NaCl, 10 mM Tris-HCl pH 7.4, 0.05% Tween-20) for 1 h at room temperature (RT). The membrane was incubated with primary antibodies for β-catenin (Cell Signalling, 1:2000 dilution), total GSK-3α/β (D75D3) (Cell Signalling, 1:1000 dilution), phospho-GSK3β (Ser9) (Cell Signalling, 1:1000 dilution), PS1 (Abcam, 1:1500 dilution), phospho-BRCA1(Ser1524) (Cell Signalling, 1:1000 dilution), and γH2AX (Cell Signalling, 1:2000 dilution) in 5% non-fat dry milk in TBST overnight at 4 °C. Then the membrane was washed, probed with HRP-conjugated secondary antibodies (Bio-Rad) at a dilution of 1:20,000 for 2 h at RT, and incubated in chemiluminescent reagents (Bio-Rad), and Primax blue-sensitive film was used for detection. The film was analyzed by densitometry, with GAPDH used for loading standardization.

Immunoprecipitation. Cells were lysed in the RIPA buffer. Agarose beads with protein G (Invitrogen, cat. no. 15920-010) were washed twice with PBS, blocked with 5% BSA in PBS for 2 h at 4 °C, and centrifuged at 14,000× *g* for 1 min at 4 °C. The extracts containing 0.5 mg protein were pre-cleared with agarose beads with protein G, centrifuged at 14,000× *g* for 1 min at 4 °C, and incubated for 2 h at 4 °C with mouse monoclonal anti-PS1-FL antibody (Santa Cruz). Aliquots of 40 µL of protein G-agarose beads were added to the lysates, incubated overnight at 4 °C, and centrifuged at 14,000× *g* for 5 min at 4 °C. Pellets and supernatants were boiled for 5 min at 95 °C in Laemmli buffer, subjected to SDS-PAGE and immunoblotted as above for β-catenin, total GSK3, GSK3β(PSer9), and APP.

Immunocytochemistry and colocalization assay. Fibroblasts cultured on glass coverslips were incubated for 10 min in 50 mM NH_4_Cl in PBS at RT, permeabilized with ice-cold 0.1% Triton X-100 in PBS for 5 min, and blocked with 5% fetal bovine serum in PBST (PBS with 0.05% Tween-20) for 30 min. The coverslips were then incubated for 1 h at RT with antibodies against β-catenin, GSK3β(PSer9), or β-amyloid. After 5 washings for 5 min each with PBST, the coverslips were incubated with secondary antibodies for 1 h at RT. Alexa Fluor^®^-488-conjugated goat anti-mouse (cat. no. A-11001) and Alexa Fluor^®^-647-conjugated chicken anti-goat antibody (cat. no. A-21469) (Thermo Fisher Scientific, Invitrogen) were used at a 1:1000 dilution. Visualization was performed using a Zeiss 780 microscope at the Laboratory of Advanced Microscopy Techniques of the Mossakowski Medical Research Centre. Colocalization was quantified with the Fiji J software version 1.52 with the Coloc 2 module.

Statistical analysis. Bar graphs or data points are presented as the means ± SEM of n observations (stated in the figure legend). Statistical significance was determined at the 95% confidence level by a one-sided Student’s *t*-test for either unpaired or paired samples or a one-way ANOVA (stated in the figure legend), with the differences considered statistically significant at * *p* < 0.05 or ** *p* < 0.01.

### 2.3. Results

#### 2.3.1. Patients Genetic Background

In a female patient with a clinical diagnosis of early-onset Alzheimer’s disease, a non-synonymous heterozygous transition G > A in *PSEN1* (NM_000021.4;c.1078G > A), resulting in an amino acid substitution of alanine at the 360 position by threonine (Ala360Thr), located in the large cytosolic loop of PS1, was identified (Figure 1). Although the large cytosolic loop of PS1 is challenging for classical modeling, several assumptions can be made about the structural consequences of the A360T mutation. Amino acid change from alanine to threonine may promote destabilization of beta helix structure located between E356 and L369 and facilitate the formation of small bundles, preferentially alpha helical shapes of distinct properties than beta helix. The location of the A360 residue in the protein structure of PS1 is depicted based on the Alphafold model, depicted in the top panel, and shown in a non-mutated right-bottom and mutated protein structures in the left-bottom top model generated by the DDMut Tool (Figure 1C). The mutation was reported for the first time in a sporadic, early-onset case [3]. We studied the impact of the A360T substitution on the pathological processes observed in Alzheimer’s disease using patient-derived primary fibroblasts. Access to the family history was limited, and only samples from the proband and her mother could be genotyped for presenilin 1, presenilin 2, and APP mutations; the only mutation found was the A360T in *PSEN1* in the proband. The genotyping data were in agreement with clinical observations of the early-onset AD phenotype in the proband and no neurological symptoms in her mother, who died at the age of 81 due to cardiovascular insufficiency and atherosclerosis. The pedigree of the studied A360T families is provided in Figure 1D.

#### 2.3.2. Genetic Profiling and In Silico Phenotyping by Whole Exome Sequencing 

We performed WES to validate the A360T mutation and identify additional possible genetic variants and modifiers contributing to the complex genetic architecture of the disease. Whole exome sequencing was performed for three patients: patient 5241 (*PSEN1* mutated), patient 7676 (non-mutated), and patient 7677 (*PSEN1* mutated, father of 7676). The pedigree of the studied A360T families is provided in Figure 1D. The patient 5241 is the one whose fibroblast cell line was available, and this patient was the base for further analyses for the majority of WES processing together with the 7677 case. Multi-step data processing was carried out, including quality control, trimming of raw reads, mapping of the trimmed reads to a reference genome, variant calling, annotation, and prioritization by the Exomiser tool. Basic analyses by Exomiser tool of WES data provided on the pathogenicity and frequency of the A360T mutation in PSEN1. The frequency of the variant was provided from various databases and was as follows: ExAC AMR: 0.0086%, ExAC NFE: 0.0060%, gnomAD_E_AMR: 0.0030%, gnomAD_E_NFE: 0.0063%, and gnomAD_G_NFE: 0.0067%. These data pertain to the frequency of the A360T variant in different populations. Next to that, the pathogenicity of the A360T variant was as follows: Polyphen2: 0.182, Mutation Taster: 1.000, SIFT: 0.25, CADD: 0.995, REVEL: 0.522, MVP: 0.850. 

Overall, the resulting list of genetic variants was subjected to functional analyses for associated signaling pathways, distinguishing between coding and non-coding sequences and variants. WES analysis confirmed Sanger sequencing results for the *PSEN1* A360T variant in patients 5241 and 7677 and its absence in the patient 7676 case. The Exomiser analysis in the HPO mode ranked the A360T variant with the highest priority #RANK score, indicating its involvement in the causativeness for AD. Also, in the noHPO mode, A360T ranked high at position 35 out of more than a thousand variants.

Based on Exomiser data for the A360T variant, the frequency ranged between ExAC AMR: 0.0086%, ExAC NFE: 0.0060%, gnomAD_E_AMR: 0.0030%, gnomAD_E_NFE: 0.0063%, and gnomAD_G_NFE: 0.0067%. The A360T variant appears to be relatively rare in the populations mentioned. Its frequencies, ranging from 0.0030% to 0.0086%, indicate that this genetic mutation is not common in these specific groups. In a broader context, a mutation with a frequency below 1% is often considered rare. Next to the frequency data, the pathogenicity of the a360T variant was as follows: Polyphen2: 0.182, Mutation Taster: 1.000, SIFT: 0.25, CADD: 0.995, REVEL: 0.522, MVP: 0.850. The pathogenicity data consists of scores from various computational tools used to predict the potential impact of a genetic variant like A360T on protein function and, by extension, its association with diseases. A score of 0.182 suggests that the Polyphen2 tool considers the A360T variant to be benign or unlikely to be damaging to the protein’s function. Whereas Mutation Taster, with a score of 1.000, classifies the A360T variant as disease-causing. SIFT at 0.25 predicts the variant to be neutral, while CADD at 0.995 indicates a very high likelihood that the A360T variant is deleterious. Similarly, the REVEL score of 0.522 suggests a moderate likelihood that the A360T variant is pathogenic. Interpreting these scores collectively, the A360T variant appears to have conflicting predictions. While Mutation Taster, CADD, REVEL, and MVP suggest a higher likelihood of pathogenicity, Polyphen2 and SIFT provide conflicting or inconclusive results. It is important to note that computational tools have their limitations, and experimental validation is typically required to confirm the actual impact of a variant on protein function and its association with diseases. Given the conflicting nature of the predictions, further functional studies and clinical observations would be necessary to definitively assess the pathogenicity of the A360T variant. Researchers and clinicians often consider multiple lines of evidence, including computational predictions, population frequency data, and functional assays, to make comprehensive and accurate assessments of a variant’s pathogenicity.

Since the consequences of the A360T substitution in the cytosolic loop of presenilin are still under discussion as to its pathogenicity (benign or hazardous) and multigenic component cannot be excluded, we analyzed all the variants found in the patients with respect to their possible input into AD. The variants were prioritized and annotated by Exomiser in the HPO or noHPO mode, followed by functional analyses using Pathview, GAGE, Manhattan (qqman), and biomaRt packages from the R environments, as well as the online tools WebGestalt (WEB-based GEne SeT AnaLysis Toolkit) and EnrichR. 

The 5241 patient carried 61,437 SNPs annotated to the reference genome by SNPeff software version 5.2 by the SNPsift too, of which 61,370 SNPs were assigned to the Ensembl database with the Biomart R package based on chromosome position. This dataset was combined with RNAseq expression data comprising 59,794 IDs of expressed genes to give a WES-RNAseq dataset of 47,771 unique IDs. The full dataset with the top 12 significant expression quantitative loci (eQTL) specific to 5241 cases carrying the A360T mutation is shown in the Manhattan plot (Figure 2A). Following advanced variant filtering, annotation (by SNPeff and the Exomiser tool), and Exomiser prioritization, 1174 variants were called in patient 5241 (892 non-coding and 282 coding ones), 1152 variants in patient 7676 (1105 non-coding and 248 coding ones), and 1190 variants in patient 7677 (955 non-coding and 235 coding ones) (Appendix A). Of those, 24 SNPs were common to patients 5241 and 7677, among which 19 SNPs were represented in the RNAseq data, 643 of which gene expression was detected by RNAseq uniquely to the A360T case (Figure 2B,C, Appendix A). 

The 19 hits of WES-RNAseq data present in the A360T female patient were (listed in order of rising *p*-value): NAPG, IRF2, SPTB, PRKCA, OTOGL, MUC12, MUC12, ERP44, PSEN1, STMN4, TRRAP, TSPEAR, MUC5AC, HPS1, RAD51B, MRPS14, PDS5A, IGHV1OR21-1, and SYNE2. Among them, PSEN1 is the most related to Alzheimer’s disease, followed by IRF2, which encodes interferon regulatory factor 2. IRF2 takes part in regulating immune responses and has been implicated in neuroinflammation, a key component of AD pathology. However, while IRF2 has been implicated in AD, its relevance to the disease and the mechanism of action are still being investigated. In our RNAseq dataset, the whole IRF family has been found (Appendix A).

Overall, the IRF family was downregulated in the A360T fibroblast of female proband, and *IRF2* downregulation was statistically significant. This is consistent with an earlier report of a significantly downregulated expression of genes, including IRF2, involved in limiting IFN production and/or receptor signaling in AD patients, additionally showing that the type I interferon response drives neuroinflammation and synapse loss in AD [9].

The number-one most significant SNP was located in *NAPG* (compare Appendix A). There is no specific link between the *NAPG* and neurodegeneration; however, its function is critical to brain cognitive functions. The *NAPG* encodes gamma-soluble N-ethylmaleimide-sensitive factor attachment protein (gamma-SNAP), which is involved in the regulation of neurotransmitter release in the brain. Disruptions in vesicle trafficking and neurotransmitter release can lead to impaired synaptic transmission and neuronal dysfunction. Gamma-SNAP, along with beta-SNAP and synaptotagmin I, are significantly reduced in the temporal cortex of AD patients. The findings went along with impaired synaptogenesis in the AD brain [10].

*NAPG* and *IRF2* were subjected to enrichment analysis by EnrichR, giving a list of pathways (enriched in the A360T case) and biological processes directly associated with neuron degeneration: pyroptosis—a form of programmed cell death involving the release of pro-inflammatory molecules in neurodegeneration; necrosis—a cell death during neuronal loss; apoptosis—programmed cell death implicated in various neurodegenerative disorders, including AD; as well as interferon alpha and gamma response—activated during neuroinflammation. *IRF2* and *NAPG* enrichment are also related to extracellular and intracellular trafficking: intra-Golgi traffic, retrograde transport at trans-Golgi network, COPII-mediated vesicle transport, vesicle fusion, and protein secretion. This, again, suggests an involvement of dysfunctional synaptic transmission in AD. All the pathways discussed above are visualized in a clustergram (Figure 3) and listed in a table (Appendix A). 

Following Exomiser analysis of WES data without defining HPO terms related to AD or neurodegeneration, we found 42 genes with SNPs of significant score present in both 5241 and 7677 patients (A360T-carriers) and absent in 7676, a daughter of the 7677 case (Figure 4A). The indicated 42 genes carried 64 SNPs (Appendix A). Functional in silico prediction for the 42 genes revealed a list of altered pathways and processes, among which the top 18 were related to neurodegeneration in A360T carriers (Appendix A). Among 42 hits of genes with SNP alterations specific to A360T case, 8 genes had an altered expression in patient 5241 (*PRKCH*, *ARHGEF10L*, *NAPG*, *NMNAT2*, *ABCA3*, *IRF2*, *AKAP13*, and *PCSK5*). *PCSK5* was unique for case with the A360T variant, as compared to the transcriptomes of other fEOAD cases carrying other variants in *PSEN1*, reported before [8]. PCSK5 is a convertase involved in the processing of pro-beta-NGF (beta-nerve growth factor) to mature beta-NGF, a critical step in the maturation and function of the nervous system. PCSK5 has been suggested to be involved in the pathogenesis of neurodegeneration. One of the proposed mechanisms linking PCSK5 to neurodegeneration involves the processing of proteins that are engaged in the formation of abnormal protein aggregates, e.g., protein tau. It has also been shown that PCSK5 can modulate the activation of microglia, playing a critical role in neuroinflammation. The exact role of PCSK5 in neurodegeneration is deserving of further studies. In the 42-gene dataset exclusive to the A360T mutation, the associated variants were located in genes enriched in the Wnt signaling pathway and metabolic pathways (WEbGestaldt, Appendix A). Based on the analysis performed by the Pathview R package, several pathways were found to be enriched in A360T cases, i.e., calcium homeostasis, focal adhesion, MAP kinases, and the regulation of actin cytoskeleton (Appendix A, Figure 5). The indicated enriched processes are connected. Wnt ligands bind to Frizzled receptors and co-receptors, causing the release of intracellular calcium ions, activating calcium-responsive effectors, and modulating downstream signaling events. Wnt regulates several components of the focal adhesion complex, including integrins, focal adhesion kinase (FAK), and paxillin. Wnt signaling also intersects with the MAPK signaling pathway, particularly its extracellular signal-regulated kinase (ERK) arm, resulting in the downstream activation of target genes involved in cell proliferation, differentiation, and survival. Wnt also influences actin-binding proteins, which controls cytoskeletal dynamics. Overall, the links between Wnt signaling, calcium signaling, focal adhesion, MAPK signaling, and regulation of the actin cytoskeleton are complex and context-dependent. Moreover, calcium signaling, MAPK, focal adhesion, and actin cytoskeleton remodeling are all linked to GSK3β signaling. Thus, the involvement of calcium signaling, focal adhesion, MAPK signaling, and actin cytoskeleton regulation in AD pathogenesis is based on specific mechanisms and interactions that are still under investigation. It is well established that dysregulation of any of these cellular processes can contribute to synaptic dysfunction, neuronal loss, and cognitive deficits observed in AD. Understanding the intricate links among all these processes should provide novel insights into the development of potential therapeutic strategies for AD. An Exomiser-based prioritization of the WES results was also performed in the HPO mode (defined as related to neurodegenerative disorders); to avoid a systematic bias, nop-specific HPO terms were predefined. As a result, we found 252 SNP IDs in 91 genes in patient 5241, 254 SNP IDs in 90 genes in patient 7676, and 216 SNP IDs in 67 genes in patient 7677 (Appendix A). Of those, four genes (*PSEN1*, *PRKCA*, *SEL1L2*, and *MRPS14*) found in patients 5241 and 7677 but not in 7676 were shared with the noHPO set detailed above. Their expression was not altered in the patients; therefore, we assumed that their possible input into AD was due to the altered functionality of the encoded proteins. These four genes were related to Wnt signaling, neurodegeneration, gamma secretase, disinhibition of SNARE formation, and MAPK signalling, all processes clearly relevant to the disease. In addition to SNPs, the WES results also highlighted diverse indels in patients exomes. In the raw data from the A360T case, there were 39889 such entries, and after Exomiser filtering and prioritization in the HPO and noHPO modes, 3285 hits and 353 hits were obtained, respectively. Upon several filtering steps, 18 indels were found to be present in RNAseq data, present in both 5241 and 7677 patients, and common for noHPO and HPO modes (Figure 6). These 18 indels were enriched in the Wnt signaling pathway and maintained a pluripotency state (defined by EnrichR). Among the 18 indels specific to the studied A360T AD cases, the top 4 in terms of statistical significance (Appendix A) were related to neurodegeneration: *DHCR7*, *GCDH*, *MAP1LC3C*, *REV1* (Figure 7). 

#### 2.3.3. A360T Fibroblast Transcriptome versus Controls

The A360T female proband was characterized by 2321 differentially expressed genes (DEGs) compared to the transcriptome of healthy controls (Figure 8A, Appendix A) and 528 DEGs compared to the transcriptome of other fEOAD subjects carrying different mutations in *PSEN1* (Figure 9A, Appendix A). The majority of DEGs were down-regulated in A360T cells, compared to controls (Figure 8B) and fEOAD cells (Figure 9B). The PCA plot was used to visualize transcriptomic distance and relatedness between different *PSEN1* genotypes and between different controls and A360T cells (Figure 8C and Figure 9C). A clinical description of fEOAD subjects was already published [8] and can be found in the Appendix A.

The Ingenuity Pathway Analysis (IPA) revealed that the A360T patient’s transcriptome showed disturbances in the cell cycle checkpoint (CCC) and DNA damage response (DDR). On the other hand, the Wnt-related genetic network constituted 6% of all of the enriched signaling pathways provided by IPA in the A360T patient, compared to only 1% in the other fEOAD patients (Figure 10A). This analysis also demonstrated other pathways that could be involved in AD pathology in the A360T case, indicating promising targets for future studies. According to the IPA, the Wnt/β-catenin/GSK3β signaling pathway was found to be inhibited in A360T cells, with a Z-score of −2.236 at a *p*-value of 0.015. Additionally, the genes *WNT2*, *FZD1*, *TCF4*, *SMO*, *WNT4*, *TCF7L2*, and *WNT9A* were identified as enriched and are depicted with blue arrows (Figure 10B). Furthermore, the translocation of β-catenin to the nucleus was predicted to be enhanced, as depicted in orange in Figure 10B (see Appendix A). Several *WNT* genes were differentially expressed in A360T patients compared with control fibroblasts: *WNT5A*, *WNT4*, *WNT2*, and *WNT9A* were downregulated, and *WNT10B* was upregulated. Thus, the predicted outcome of Wnt signaling comprised inhibition of cell growth, disrupted actin filament organization, and upregulated cell adhesion.

#### 2.3.4. Wnt/β-Catenin Signaling in Cell Adhesion and Cytoskeleton Remodeling in the Light of the Senescent Morphology of A360T Fibroblasts

Wnt/β-catenin signaling plays a critical role in regulating cell morphology and cell attachment, with stabilized β-catenin activating the transcription of genes involved in cell adhesion and cytoskeleton remodeling [11]. Therefore, we have established a substantial modulation of the Wnt/β-catenin pathway in A360T fibroblasts. Unlike the control fibroblast and those derived from fEOAD patients carrying transmembrane domain (TMD) PS1 mutations, which were all spindle-like shapes, the A360T fibroblasts had a large, flattened cell body (Figure 11A). They also exhibited increased granularity and large, irregularly shaped nuclei (Appendix A), and their proliferation rate was lower than that of the other fibroblasts, as evidenced by a higher population doubling time (Figure 11B). Notably, these features likely explain the inability of the A360T fibroblasts to induce pluripotent stem cells, which we could not achieve in as many as five attempts, while other cell lines during the same experimental batch were successfully reprogrammed in the first attempt. Overall, the senescence-like profile plays a key role in age-related diseases, such as degeneration of the nervous system, which corresponds to the A360T case.

#### 2.3.5. Unique Transcripts of A360T Fibroblasts

As described above, we compared the A360T cell line to controls and to fEOADs, which provided lists of 2321 and 528 DEGs, respectively (Appendix A). Compared with the transcriptomes of control fibroblasts and those from non-A360T fEOAD cases, the A360T fibroblasts expressed 419 unique DEGs (Figure 12A and Appendix A). Among these DEGs, 63 were upregulated and 356 were downregulated relative to all the reference transcriptomes (Figure 12B,C). Again, this set was significantly enriched in genes related to the GSK3β/β-catenin/Wnt pathway. This prompted us to study the pathway activity in the A360T fibroblasts at the cellular and biochemical levels. 

#### 2.3.6. GSK3β/β-Catenin/Wnt Signaling in A360T Fibroblasts

Consistent with the RNAseq data, we found that the A360T fibroblasts had greatly reduced levels of the active form of β-catenin in the cytosol but higher levels in the nuclei compared to both control and fEOAD fibroblasts (Figure 13A); the level of β-catenin was similar in all the reference fibroblast lines, including those with *PSEN1* mutations other than A360T. Notably, the fraction of GSK3β phosphorylated on serine 9, i.e., inactive as a kinase, was markedly higher in the A360T fibroblasts than in the control ones (Figure 13B), while all the other fEOAD fibroblasts showed significantly reduced levels of Ser9-phosphorylated GSK3β, in agreement with earlier reports. Since PS1 is known to interact with both β-catenin and GSK3β, we examined these interactions in the A360T fibroblasts by co-immunoprecipitation, again in comparison with control and fEOAD fibroblasts. The level of the complex with β-catenin was much lower (Figure 13C), and that of the complex with GSK3β(Ser9) was much higher (Figure 13D) in the A360T fibroblasts than in the other fibroblast lines examined. The subcellular localization of β-catenin and GSK3β(Ser9) in A360T fibroblasts was similar to that in controls and weaker than in other fEOAD cells (Figure 14A and Appendix A). Conversely, the signal for GSK3β(Ser9) was significantly stronger in A360T cells than in all the other cells (Figure 14B, Appendix A). As in the co-IP assays, the abundance of respective complexes as determined by signal colocalization was lower for the PS1-β-catenin one (Figure 14C) and higher for PS1-GSK3β in the A360T fibroblasts (Figure 14D). The corrected total cell fluorescence (CTCF), calculated using Fiji J software version 1.52, for GSK3β and presenilin 1 showed that the colocalization could be due to the enhanced signal of GSK3β (Appendix A). A higher content of the GSK3β(Ser9)-PS1 complex was also observed in another fEOAD case with a substitution in the cytosolic loop of PS1 (R307S), a case with a substitution in the TMD in PS1 (L424R), and one near-TMD substitution (P267L). We found a relatively weak immunocytochemical staining for Aβ in the A360T fibroblasts, whereas the L424R, P267L, and R307S cases showed a strong Aβ signal, significantly higher than in controls (Figure 15A,B). The content of both amyloid beta 1–40 and 1–42 determined by ELISA was lower in the A360T fibroblasts than in the control ones (Appendix A).

This was consistent with the abundance of APP immunoprecipitated with PS1, which was the lowest in the R307S, L424R, and P267L cells, whereas in the A360T cells it was comparable to that in the controls (Figure 15C). The bands at 100 and 140 kDa correspond to various isoforms of mature and immature APP, including APP695, APP770, and APP751, recognized by the antibody. The data shown in Figure 9C suggests that the rate of APP processing in A360T fibroblasts is close to normal and does not lead to a pathological accumulation of Aβ peptides. Based on the above findings, we propose that the aberrant relationship between PS1, β-catenin, and GSK3β in the A360T fibroblasts affects the availability of PS1. Indeed, the amount of both intact PS1 and its NFT and CTF fragments was significantly lower in the A360T fibroblasts than, on average, in the other fEOAD ones or the controls (Figure 15D).

Recently, it was found that fEOAD is associated with disturbances in the DNA damage response (DDR) and cell cycle checkpoints (CCC) with the key role of an overreactive BRCA1 [8]. Consistent with this, an inspection of the A360T transcriptome indicated a similar recruitment of the components of DDR and CCC. To verify these predictions, we determined the oxidative damage response of the A360T fibroblasts by measuring the content of γH2AX and BRCA1 phosphorylated on Ser1524 after doxorubicin treatment; neither BRCA1 activation (Figure 16A,B) nor recruitment of γH2AX was observed (Figure 16A,C) in contrast to the other fEOAD fibroblasts. The response of GSK3β and β-catenin in A360T fibroblasts to oxidative stress was also different from that in the other cells (Figure 16A,D,E). After doxorubicin treatment, the level of GSK3β(Ser9) initially dropped slightly in A360T, whereas in the other cells it initially doubled and then changed only moderately. In contrast, β-catenin abundance was unaffected in all the cells examined, with the clear exception of A360T, where it rose severalfold from an initially very low level to reach a similar abundance to that in the control.

Since the transcriptomic analysis suggests recruitment of β-catenin with a downstream up-regulation of cyclin D1 in A360T fibroblasts (Figure 10B, Appendix A), we tested cell cycle steps. The number of A360T fibroblasts in the S phase was less than half that found in the other cells studied. The number of A360T fibroblasts in G1 was only slightly higher than in controls but significantly higher than in the other fEOAD. The number of A360T fibroblasts in the G2/M phase was non-significantly lower than in controls and significantly lower than in the other fEOAD-derived fibroblasts. Notably, very few A360T cells were apoptotic, similarly to the controls and other fEOAD cases. Doxorubicin treatment did not affect the cell cycle but significantly increased the frequency of apoptosis exclusively in the A360T fibroblasts (Figure 17). This suggested a dysfunction of the β-catenin/cyclin D1 pathway in cell cycle regulation. Summing up, there was a slight inhibition of the G1 → S transition in the A360T fibroblasts. Of note, while β-catenin was found to control the G1 → S transition, cyclin D1 is required for the progression through the G1 phase and is involved in the expression of genes required for entry into the S phase [12]. 

## 3. Discussion

### 3.1. Possible Neuroprotection by Increased Phosphorylation of GSK3β in A360T Cells

GSK3 is constitutively active under basal conditions and is inhibited upon phosphorylation on serine 9 in GSK3β and serine 21 in GSK3α. GSK3β kinase increases with age and in AD pathology. Constitutively hyperactive (unphosphorylated) GSK3β is associated with the pathogenesis of sporadic and familial AD [13], whereas inhibited GSK3β phosphorylated at Ser9 is believed to be neuroprotective [14]. Overall, GSK3β promotes phosphorylation and the formation of toxic tau species, and its pro-inflammatory activation impairs LTP, neurogenesis, and memory, thus representing a good therapeutic target against AD [15].

Inhibition of GSK3β has been shown to enhance the repair of double-strand DNA breaks in hippocampal neurons [16]. The results of the present study indicate that fibroblasts from a patient with the A360T mutation in presenilin 1 have a higher level of GSK3β(Ser9) compared to normal controls and even more relative to other fEOAD cases tested. The presence of inactive GSK3β(Ser9) in A360T cells suggests potential neuroprotective actions and the induction of a DNA damage response. In contrast, other subjects with fEOAD showed an overall increase in total GSK3β content and a decrease in its phosphorylated form. Interestingly, an in vivo study showed that the content of GSK3β(Ser9) in AD patients changes during the course of the disease, with a significant increase in the early phase and a decrease in the late phase of the disease [17]. Thus, markedly enhanced GSK3β phosphorylation in patient A360T suggests its early stage of the disease.

### 3.2. The Impact of A360T-Mutated PS1 on GSK3β and β-Catenin and No-Amyloid-Related Downstream Targets 

Active GSK3β binds the PS1 region, comprising residues 250–298; some PS1 mutations have been reported to affect the binding affinity. The PS1-GSK3β interaction is enhanced in amyloid pathology [18]. However, in the A360T cells, the amount of the active non-phosphorylated GSKβ did not differ from that in normal the control, in contrast to other fEOAD cases. On the contrary, A360T-PS1 was found to immunoprecipitate with the inhibited GSK3β(Ser9). In the other cases of fEOAD, the PS1-GSK3β interaction was similar to the control level. At the same time, the amyloid staining was weaker in the A360T fibroblasts than in the other fEOAD cases. In addition, PS1 has been proposed to modulate the interactions of GSK3β with its substrates [19]. Therefore, by increasing the binding with GSK3β mutations in PS1 could alter the ability of GSK3β to phosphorylate its targets. We propose that the enhanced preferential binding of the A360T-mutated presenilin 1 with the inhibited form of GSK3β phosphorylated at Ser9 could affect further downstream processes such as phosphorylation of the targets of GSK3β, degradation of β-catenin, and β-catenin-dependent gene expression (i.e., genes related to the cell cycle and apoptosis). In addition, phosphorylation of PS1 serines 353 and 357 by GSK3β induces a structural change of the hydrophilic loop of PS1, reducing its interaction with β-catenin and causing a drop in β-catenin phosphorylation [19]. The phosphorylation of the serines in the proximity of the A360T mutation appears to be critical for the pathogenic ‘closed’ conformation of PS1 and enhancing amyloid production [20]. Furthermore, the presenilin loop region is essential for glycogen-synthase-kinase-3-β-(GSK3β)-mediated functions on motor proteins during axonal transport [21]. Notably, in our study, the A360T cells did not show marked amyloidopathy. The amount of β-catenin co-immunoprecipitating with PS1 was low in the A360T cells, suggesting stabilization of β-catenin, as also evidenced by its increased nuclear content leading to transcription of target genes, including cyclin D1. The alterations in the interplay between β-catenin and PS1 could also have implications for neurogenesis mediated by PS1 via β-catenin phosphorylation and Notch signaling [22]. A dysfunctional PS1 could impair adult neurogenesis and play a role in the cognitive deficits observed in Alzheimer’s disease [22]. Taken together, our results suggest that the A360T mutation could have adverse effects on its downstream partners, including those involved in neurogenesis and neuroprotection without enhancing amyloidogenesis.

### 3.3. A360T-Mutated PS1 Affects β-Catenin Pool and Downstream Gene Expression

It has been suggested that PS1 can destabilize the free pool of β-catenin. Under physiological conditions, PS1 promotes proteasomal degradation of β-catenin by stimulating its phosphorylation by GSK3β. In agreement, a decrease in PS1 activity has been associated with an increase in the β-catenin level [23]. On the other hand, the content of β-catenin is usually reduced in the brains of Alzheimer’s patients with PS1 mutations. The resulting loss of β-catenin signaling increases neuronal vulnerability to apoptosis. Accordingly, we showed a decreased level of active β-catenin and suggested its contribution to the non-amyloidogenic pathomechanism in the studied patient case. However, more research is needed to fully understand the molecular mechanisms underlying the relationship between PS1 activity and possible β-catenin depletion. Overall, depletion of total β-catenin makes cells more sensitive to apoptosis. Knockdown of β-catenin significantly altered the expression level of 139 genes involved in pro-apoptotic pathways [24]. Similarly, the reduction of β-catenin due to the A360T mutation in PS1 should affect β-catenin-dependent expression of genes involved in cell cycle regulation, cell death, and apoptosis. Indeed, numerous genes related to apoptosis, cell death, and cell cycle signaling, including cyclins and cyclin-dependent kinases, were dysregulated in A360T fibroblasts (Appendix A and Appendix A). β-catenin directly binds to the promoter region of *CCND1* and stimulates its expression, which promotes the G1 → S phase transition by activating cyclin-dependent kinases (CDKs) [25]. It is also known that the level of cyclin D1 needs to be high during the G1 phase for the cell to initiate DNA synthesis, but then it has to decrease during the S phase to allow efficient DNA synthesis [16]. Overall, dysregulation of cyclin D1 expression or activity can lead to abnormal DNA synthesis, which can contribute to various disease conditions, including cancer and neurodegeneration [26]. Accordingly, cyclin D1 was upregulated in A360T cells, and this was accompanied by a slight accumulation of cells at the G1 phase and a corresponding reduction in those in the S phase (see Figure 11 and Appendix A). It has also been reported that overexpression of cyclin D1 together with Cdk4 or Cdk2 prevents DNA repair [27]. Notably, in addition to cyclin D1, several Cdks and also other cyclins (A2, B1, B2, E2, and D3) were upregulated in A360T cells (Appendix A and Appendix A). In addition, activated GSK3 plays an inhibitory role in cell cycle progression and cell proliferation, i.e., by regulating the stability of cyclin E and D1, Cdc25A, and c-Myc, which are all critical for the G1 to S transition. The activated form of GSK3 promotes the degradation of both cyclin D1 and cyclin E. GSK3 activity is needed in quiescent cells and in the G1 phase and is critical for progression to the S phase. The reduced proliferation rate of the A360T cells is therefore consistent with the prevalence of the inhibited form GSK3β(Ser9) in these cells. Overall, abnormal recruitment of the key players of the cell cycle observed in the cell derived from the A360T case may suggest that the neurons of this patient could exhibit cell cycle re-entry and the induction of apoptosis. Despite the altered cell cycle and expression pattern of numerous apoptosis-related genes, the A360T fibroblasts showed a negligible frequency of apoptosis. However, it was significantly increased upon doxorubicin-induced redox stress. This increased sensitivity of A360T cells to a pro-apoptotic agent suggests that the A360T PS1 mutation could increase the rate of apoptosis of neurons in vivo in the patient’s brain. 

### 3.4. The Wnt/β-Catenin/GSKβ Axis Reveals a Dual Mechanism in A360T Cells

Regarding the inconsistency in the Wnt, β-catenin, and GSK3β status in A360T cells, it seems likely that the elevated content of GSK3β(Ser9) could be due to enhanced Akt/PIK3 signaling intersecting with the Wnt pathway [24]. We found that the expression of various *WNT* genes was downregulated in the A360T cells, which suggested the “off” state of Wnt signaling, consistent with the decreased level of active β-catenin. The decrease in active β-catenin suggested its enhanced degradation, likely with the contribution of GSK3β [28]. On the other hand, our RNAseq functional predictions as well as biochemical data demonstrated an increased translocation of β-catenin into the nuclei and its targets, and indeed an enhanced expression of β-catenin-dependent genes, such as cyclin D1. It was in turn consistent with the accumulation of inhibited GSK3β(Ser9). The low content of β-catenin in A360T cells suggests that most of it was targeted for degradation, suggesting the action of non-phosphorylated GSK3β. On the other hand, the activation of β-catenin-dependent genes, such as cyclin D1, indicates quite the opposite prevalence of a phosphorylated and inhibited form of GSK3β, which indeed was detected in abundance in the A360T cells. This apparently points to a dual mechanism in which GSK3β acts like a molecular switch seeking to rescue the A360T cells. Finally, it should be noted that β-catenin also binds to the membrane E-cadherin complex with a contribution from PS1, which is enhanced when the cytosolic loop of PS1 is altered [23]. The A360T mutation is exactly the same as the PS1 mutation.

### 3.5. GSK3β-Dependent Induction of DNA Damage Stress Response (DDR) and Cell Cycle Checkpoint Control (CCC) after Doxorubicin Treatment of the A360T Cells

A crosstalk between the DNA damage response (DDR) and PI3K/Akt/GSK3β signaling has been documented [11]. The induction of DDR with doxorubicin in A360T cells did not increase the abundance of γH2AX or phopshoBRCA1, as we found in other fEOAD cells [8]. On the other hand, similar to the other fEOAD cells, a transcriptomic analysis showed dysregulated expression of numerous components of the DDR and CCC signalling. After treatment with doxorubicin, the total level of β-catenin increased in A360T cells, but the level of GSK3β(Ser9) decreased, suggesting that GSK3β could be re-activated from its inhibition state, thereby losing its neuroprotective properties. On the other hand, the increase in β-catenin abundance could signal an induction of expression of genes related to an oxidative-stress response. At the same time, the classical DDR route remained silent in those cells. It is known that GSK3β is negatively regulated by PI3K signalling through Akt [29] and could be involved in the control of the expression of proapoptotic genes through the p53 protein [30]. The latter links DNA damage, cell cycle perturbation, and apoptosis routes. Similarly, we and our collaborators have described the phenomenon of overactivation of CCC and DDR, driven by p53 and/or p21 signaling routes, in fEOAD patients carrying *PSEN1* mutations such as P117R, P117L, I213F, or L153V [8,31,32]. In the case studied here with a cytosolic loop mutation in PS1, the GSK3β/β-catenin signaling could be activated after DNA damage independently of BRCA1, in contrast to the other fEOAD cases.

### 3.6. Does the Low Level of β-Catenin in A360T Cells Facilitates the Inhibition of GSK3β?

An underappreciated feature of GSK3β(Ser9) is that phosphorylation does not completely inhibit its activity [13]. In fact, the phosphorylated N-tail of GSK3β(Ser9) associates intramolecularly with its substrate-binding pocket, preventing the binding of the substrates of GSK3β, including β-catenin. Thus, with an increasing concentration of a substrate, it outcompetes the phospho-domain of GSK3β and may in fact undergo phosphorylation itself [23]. Accordingly, as the concentration of β-catenin increases, it will compete for binding with the phospho-domain of GSK3β preventing its phosphorylation in Ser9. However, the level of β-catenin was low, which could indirectly contribute to an increased level of GSK3β(Ser9) in A360T cells. On the other hand, the inhibitory phosphorylation of GSK3β at serine 9 does not eliminate phosphorylation of β-catenin, allowing its accumulation at a steady-state level as a substrate of GSK3β [13]. Therefore, it is possible that the low level of active β-catenin in A360T cells may be insufficient to trigger the auto-inhibitory phosphorylation of GSK3β, which in turn is the reason for the high level of GSK3β(Ser9).

### 3.7. The A360T Substitution Likely Disturbs β-Helix in the C’ Terminal Fragment of PS1

The endoproteolysis of mature PS1 at 52 kDa generates an amino-terminal fragment (NTF) (~30 kDa) and a carboxyl-terminal fragment (CTF) (~20 kDa). The N-terminal fragment of CTF, after 285 aa, is an alpha-helical one followed by a β-helix comprising E356-L369 [23], indicating the functionality of the A360T mutation. CTF interacts with modifiers of cellular adhesion (MOCA), which mediate Aβ production and cellular adhesion. Some mutations in the cytosolic loop of PS1 (e.g., D257A, D257E, D385A, and D385E) abolish holoprotein cleavage and γ-secretase activity [25]. Moreover, various mutations in PS1 or PS2 can prevent the generation of Aβ species [26]. Accordingly, we found that A360T cells had a lower amount of PS1 NFT, suggesting that this mutation reduced the splitting of PS1. If so, PS1 may be dysfunctional in this cell line, in agreement with the lower immunostaining for amyloid. Also, the transcriptome data suggest that the amyloid processing network is inhibited in A360T cells, which was not observed in other fEOAD cells.

### 3.8. Does the Wnt/β-Catenin/GSK3β Signaling Is the Only One Possible in A360T Cells? Implications of the WES Study

It is generally believed that the presence of a mutation in presenilin closes the case as far as the genetics and pathomechanisms of AD are concerned. However, this is an oversimplification; for some changes in presenilin 1, there is still conflicting anecdotal evidence about their harmfulness. Such changes certainly include changes lying in the large cytosolic loop. In this case, the WES study meets the need to determine the significance of the variant and possibly indicate potential accompanying causative processes for the disease. Overall, the significance of Wnt has been confirmed. In addition, gene candidates such as *IRF2*, *NAPG*, and *PCSK5* were defined as the critical potential variants in AD. The implications of these genes in AD have been commented on in the Section 2.3.

### 3.9. Limitations of the Study

It should be noted that this study addressing Alzheimer’s disease used skin fibroblasts as a research model and blood cells for whole-exome sequencing. Although the fibroblasts used in this study were genetically unaltered and derived directly from the patient, they have some limitations in neurobiology. The primary limitation is that, unlike neurons, fibroblasts are non-excitable and slowly proliferating cells. That drawback notwithstanding, numerous studies have utilized fibroblasts due to their preserved genotypes and potential for longitudinal phenotyping functional studies while kept in culture for extended periods.

## 4. Conclusions

In this study, we show that in fEOAD patient-derived fibroblasts with a heterozygotic A360T mutation in presenilin 1, the Wnt/β-catenin/GSK3β signaling is compromised without an accompanying overproduction of β-amyloid. Considering the fact that Wnt signaling is commonly associated with enhanced accumulation of Aβ, this report presents novel insight into AD research. In the A360T PS1 mutated fibroblasts, GSK3β was highly phosphorylated at Ser9, i.e., inactive as a kinase, which further promoted translocation of β-catenin to the nuclei. Our report provides novel data on the functional implications at the cellular and biochemical level of the A360T substitution in the large cytosolic loop of PS1 and thereby fills the gap on the significance of the less understood portion of PS1, considered to date less critical in the development of AD. We also show that *PSEN1* mutations affecting different parts of protein PS1 can present strikingly different molecular phenotypes, especially when comparing mutations located in the cytosolic loop and mutations located in the transmembrane domains of PS1. The WES data reinforced the importance of the A360T substitution as the most important and top-prioritized in the studied AD case. Our studies confirmed the role of the A360T variant and proposed several accompanying variants and processes emerging based on a multigenetic background, already postulated as being as important in the pathogenesis of AD as a single pathogen (single mutation). The results suggest that mutations located in the large PS1 cytosolic loop may be deleterious via pathomechanisms not related to amyloid processing and are focused around the Wnt/β-catenin/GSK3β signal. 

### Data Availability

RNA sequencing data have been deposited in the Sequence Read Archive NCBI NIH database as BioProject PRJNA382346. Additional data are available upon request.

## Figures and Tables

**Figure 1 ijms-24-16999-f001:**
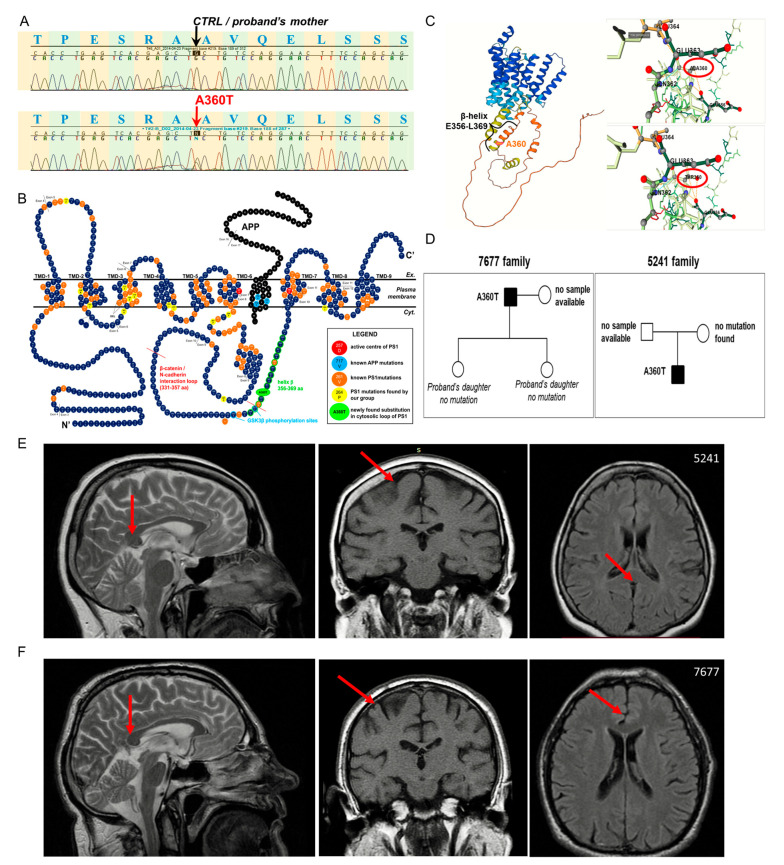
Mutation in *PSEN1*. Non-synonymous heterozygous *PSEN1* variant c.1078G>A, cDNA.1350G>A, and g.75474G>A was detected by Sanger sequencing (**A**). The mutation resulted in the substitution of alanine at position 360 by threonine (A360T), localizing in the large cytosolic loop of presenilin 1 (PS1) in the C’ portion of the protein (**B**). The N-terminal fragment (NTF) and C-terminal fragment (CTF) are generated by two cleavages of full-length PS1, first at 292 and second between 298 and 299 residues. The beta-helix, incorporating the A360T substitution, ranges from 356 aa to 369 aa, which are marked in red circles. Structural analyses of the A360T variant based on the alphafold model by the DDMut tool reveal a potent shift from beta-helix to alpha-helical bundles (**C**). The pedigrees of the analyzed families (**D**). The MRI scan of the head of 5241 female patient taken at the disease onset at age 57 representing mild atrophy (**E**). The MRI scan of the head of 7677 male patient taken at the disease onset at age 62 representing mild atrophy (**F**). Arrows indicates sites of mild atrophy.

**Figure 2 ijms-24-16999-f002:**
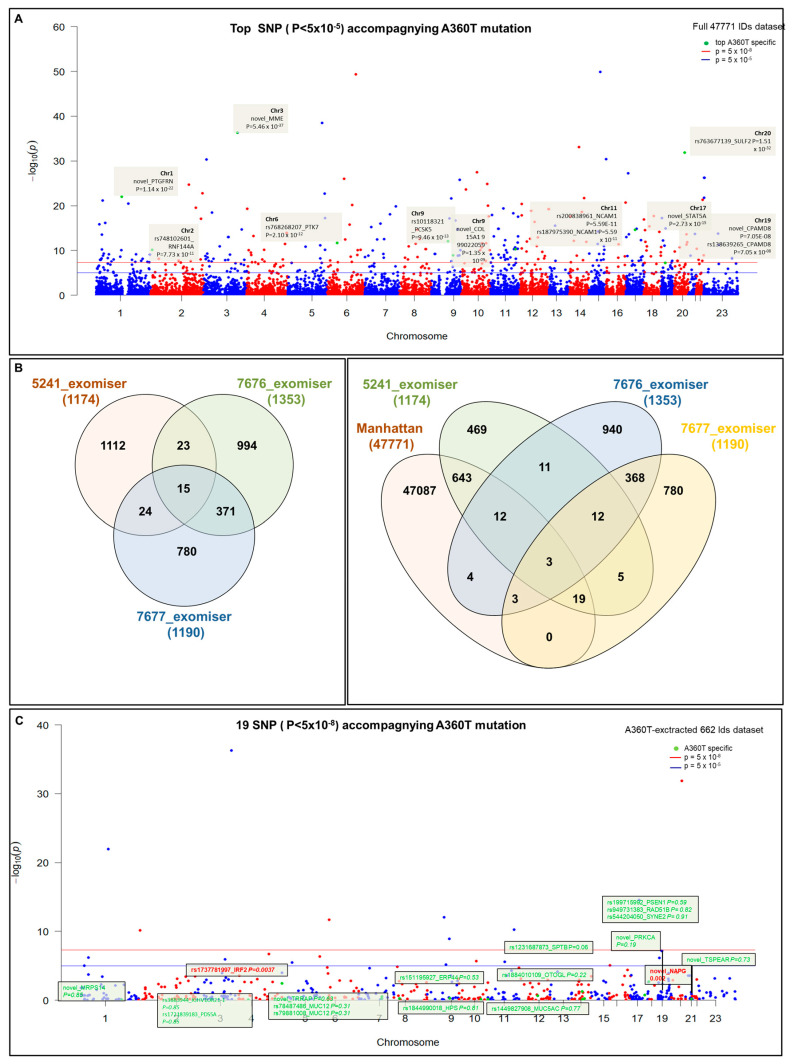
Manhattan analysis of a full dataset from WES and 47,771 RNAseq IDs. Top 12 most significant eQTLs (in terms of changes in expression) specific to patient 5241 case carrying the A360T mutation (**A**) Distribution of genetic variants among patient’s Exomiser prioritization under noHPO mode indicating 24 SNP IDs shared by patients for 5241 and 7677 cases, among which 19 SNPs were present in RNAseq data (Manhattan) (**B**). Manhattan plot of 643 SNPs, 5241-specific SNPs, and 19 SNPs defined in (**B**,**C**).

**Figure 3 ijms-24-16999-f003:**
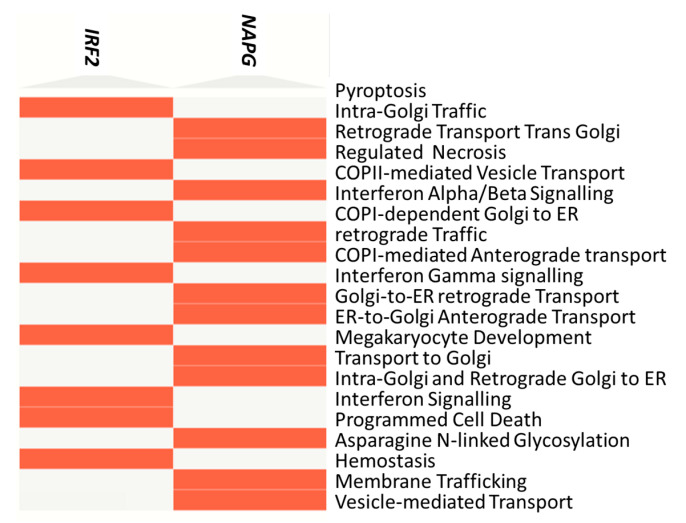
Clustergram for NAPG and IRF2 genes for 5241 cases provided by EnrichR tool.

**Figure 4 ijms-24-16999-f004:**
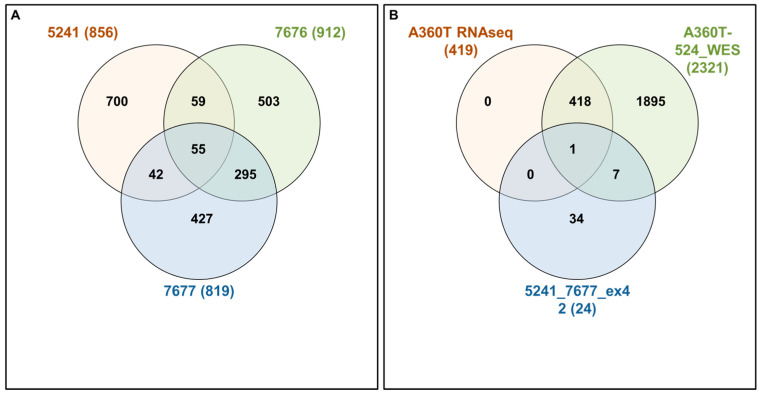
Interactivenn analysis of the lists of genes selected by Exomiser noHPO mode for 5241, 7676, and 7677 cases (**A**) and the Interactivenn comparison of preselected 42 genes (**B**) against 2321 genes differentially expressed in A3360T fibroblasts against controls and against 419 genes differentially expressed in A360T fibroblasts against other fEOAD cases.

**Figure 5 ijms-24-16999-f005:**
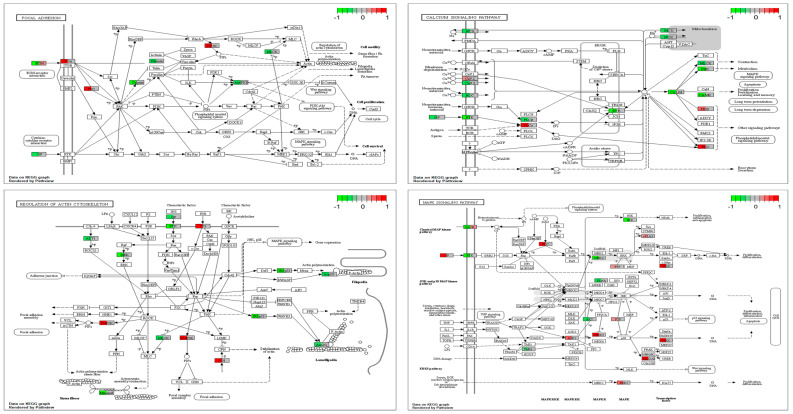
Four top enriched pathways related to SNPs shared by patients 5241 and 7677 were selected using the Pathview R package based on Exomiser noHPO results: calcium homeostasis, focal adhesion, MAP kinases, and regulation of actin cytoskeleton.

**Figure 6 ijms-24-16999-f006:**
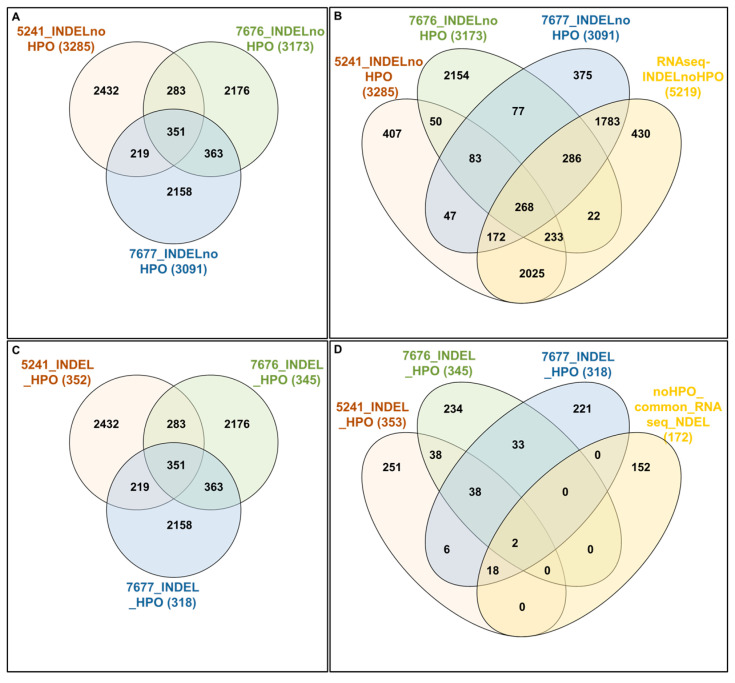
The interactive analysis of indels. Lists of indels in noHPOmode (RS_SNP_GENE) for 5241, 7676, and 7677 cases (**A**) compared with Manhattan data—full dataset of 5219 INDELS found in RNAseq data (**B**). Lists of indels in HPOmode (RS_SNP_GENE) for 5241, 7676, and 7677 cases (**C**) compared with Manhattan data—a dataset of 172 genes in RNAseq data and containing 372 INDELs and common for 5241 and 7677 cases (**D**).

**Figure 7 ijms-24-16999-f007:**
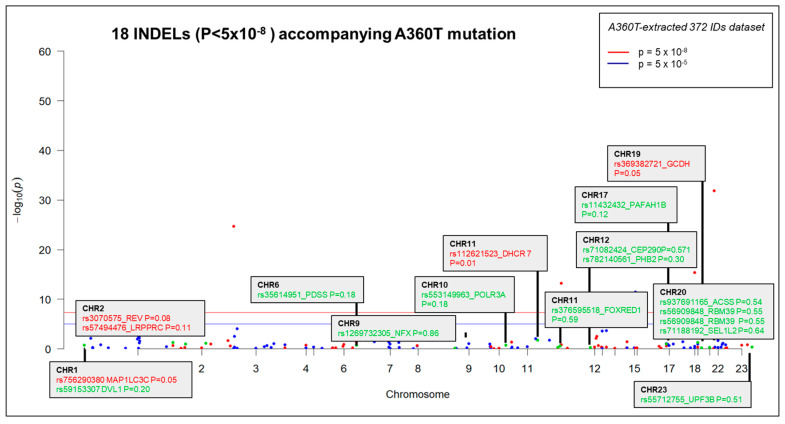
Manhattan plot of the dataset of 372 INDELS common for 5241 and 7677, with the top significant 18 eQTL specific to the 5241 case carrying the A360T mutation.

**Figure 8 ijms-24-16999-f008:**
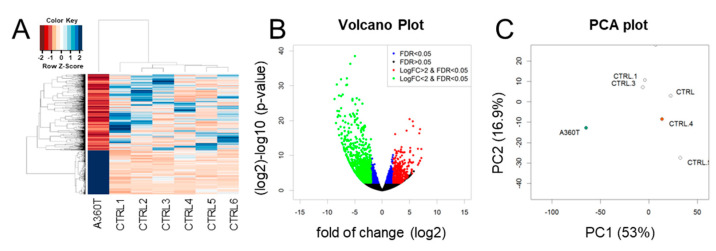
Differentially expressed genes in A360T fibroblasts compared with controls. A DESeq2 comparative analysis of the A360T transcriptome versus control ones (*n* = 6) revealed 2321 differentially expressed genes (DEGs) at fold change ≥ |2|, FDR ≤ 5%, *p* ≤ 0.05 (**A**). Downregulated DEGs were more numerous than upregulated ones (**B**). Principal component analysis showed separate clustering of A360T transcripts and controls (**C**). For comparison with other fEOAD, the fibroblasts were derived from the patients with the following mutations: AD—I211M_mother, AD.1—I211M_son, AD.2—L153V, AD.3—L424R_2, AD.4—P267L, AD.5—R307S, and AD.6—L424R_1.

**Figure 9 ijms-24-16999-f009:**
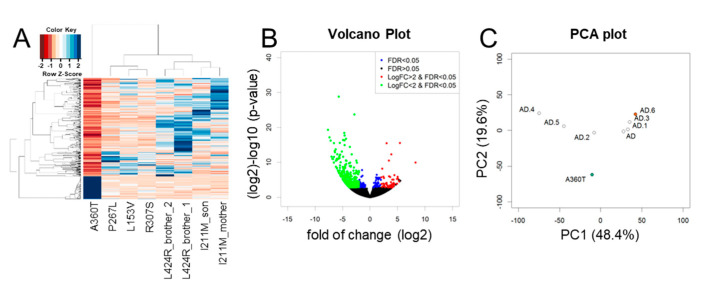
Differentially expressed genes in whole-transcriptome of A360T cells compared with fEOADs. A DESeq2 comparative analysis of the A360T case and the fEOADs group (*n* = 7) revealed 528 differentially expressed genes (DEGs) at fold change ≥ |2|, FDR ≤ 5%, *p* ≤ 0.05, as presented in the heatmap of regularized log-transformation counts (**A**). Downregulated DEGs were more numerous than upregulated ones (**B**). The principal component analysis showed separate clustering of A360T and fEOADs fibroblasts (**C**). The AD genotypes in the PCA plot are according to the given order: AD—I211M_mother, AD.1—I211M_son, AD.2—L153V, AD.3—L424R_2, AD.4—P267L, AD.5—R307S, AD.6—L424R_1.

**Figure 10 ijms-24-16999-f010:**
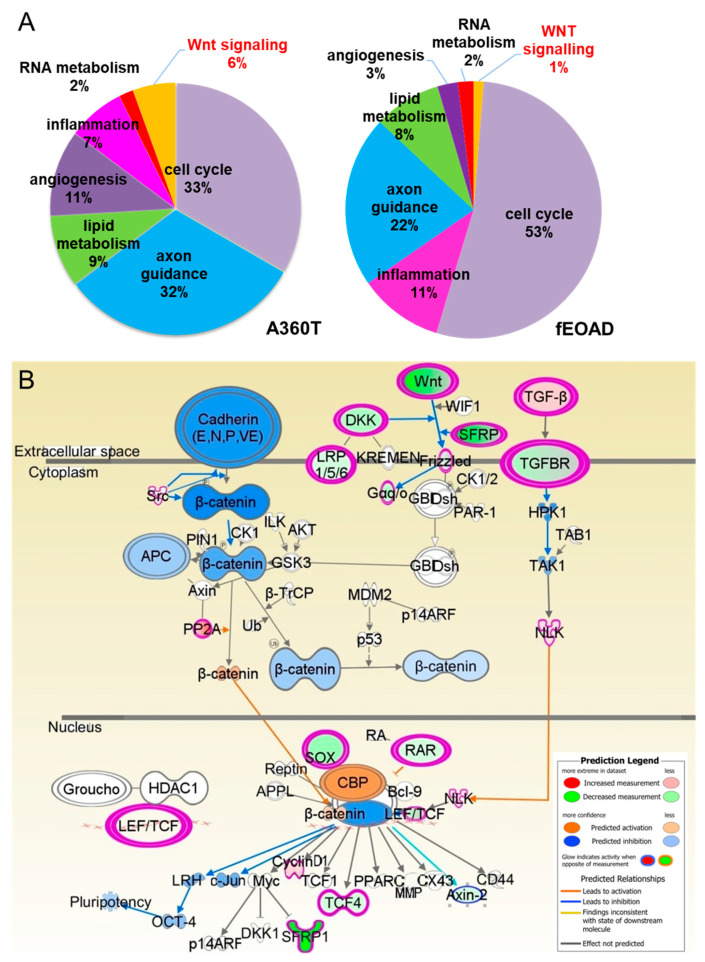
Functional enrichment analysis of differentially expressed genes in A360T fibroblast and other fEOAD cases. The DEG sets were assigned to gene ontology biological terms using the IPA tool for the A360T case (right) and other fEOADs cases (left) (**A**). Based on IPA analysis, downregulation of several *WNT* genes (in green) in A360T cells results in predicted in silico inhibition (in blue) of cytosolic Wnt/β-catenin/GSK3β signaling, accompanied by activated (in orange) nuclear translocation of β-catenin followed by modulation of its downstream genes, including upregulation of cyclin D1 (in pink) (**B**).

**Figure 11 ijms-24-16999-f011:**
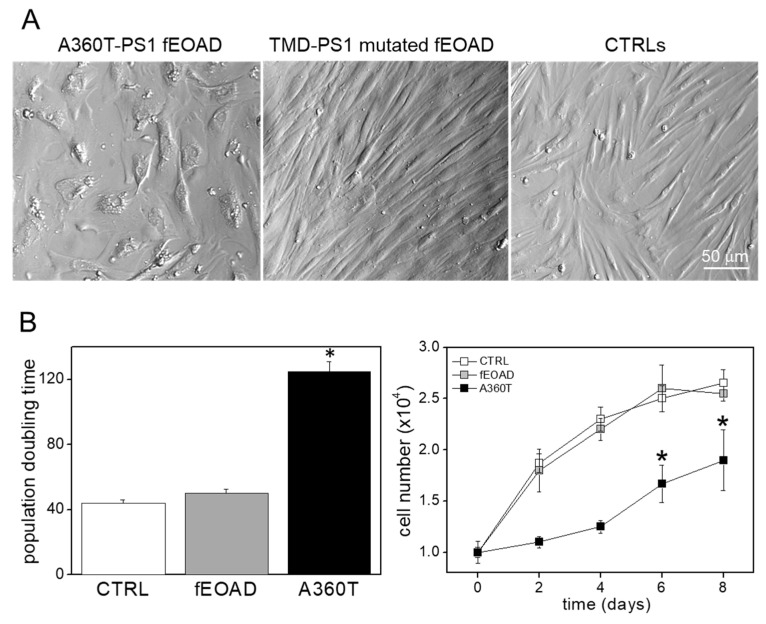
Morphology and proliferation rate of A360T fibroblasts. A360T fibroblasts had large flattened cell bodies rather than typical spindle-shaped ones, as observed for healthy controls and other fEOAD cases with mutations in transmembrane domains (TMD) of PS1 (**A**). A360T fibroblasts had a lower proliferation rate than other cells, as expressed by their higher population doubling time (*n* = 5) * *p* < 0.05 (**B**).

**Figure 12 ijms-24-16999-f012:**
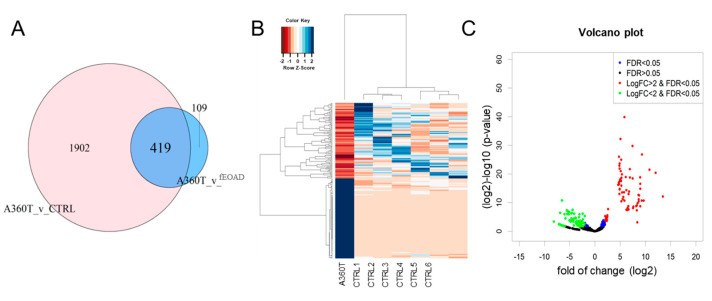
Differentially expressed genes unique to A360T fibroblasts. Genes up- or downregulated in A360T cells compared with healthy controls and other fEOAD cases were identified by comparing DEGs versus controls and fEOADs using GeneVenn tool (**A**). The 419 A360T-specific DEGs comprised 63 upregulated genes and 356 downregulated genes (**B**,**C**).

**Figure 13 ijms-24-16999-f013:**
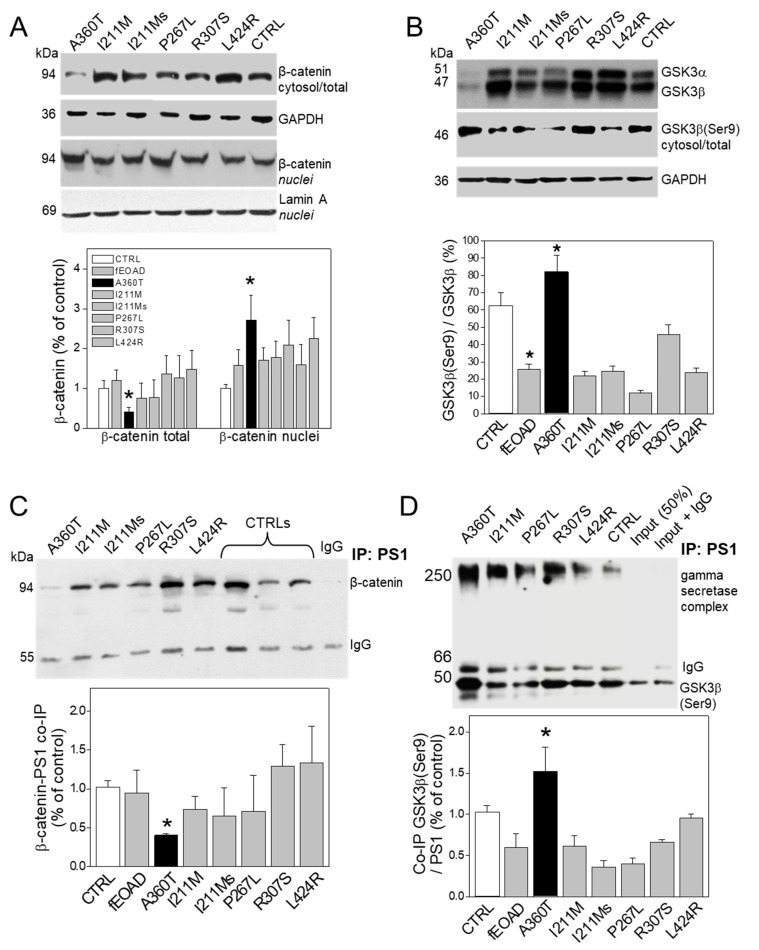
Status of β-catenin and GSK3β and their complexes with PS1 in A360T fibroblasts. The content of active non-phosphorylated β-catenin was decreased in A360T cells compared to healthy controls or other fEOAD cells in total cell lysate but elevated in nuclear fraction (*n* = 3, * *p* < 0.05) (**A**). The proportion of GSK3β phosphorylated on serine 9, standardized to non-phosphorylated GSK3β, was substantially higher in A360T cells compared with healthy controls or other fEOAD cells (*n* = 3, * *p* < 0.05) (**B**). The amount of β-catenin and GSK3β forming a complex with PS1 was lower in A360T cells than healthy controls or other fEOAD cells, *n* = 3, * *p* < 0.05 (**C**). Conversely, the amount of the PS1-GSK3β(Ser9) complex was increased in A360T cells relative to that in healthy controls or other fEOAD cells (*n* = 3, * *p* < 0.05) (**D**).

**Figure 14 ijms-24-16999-f014:**
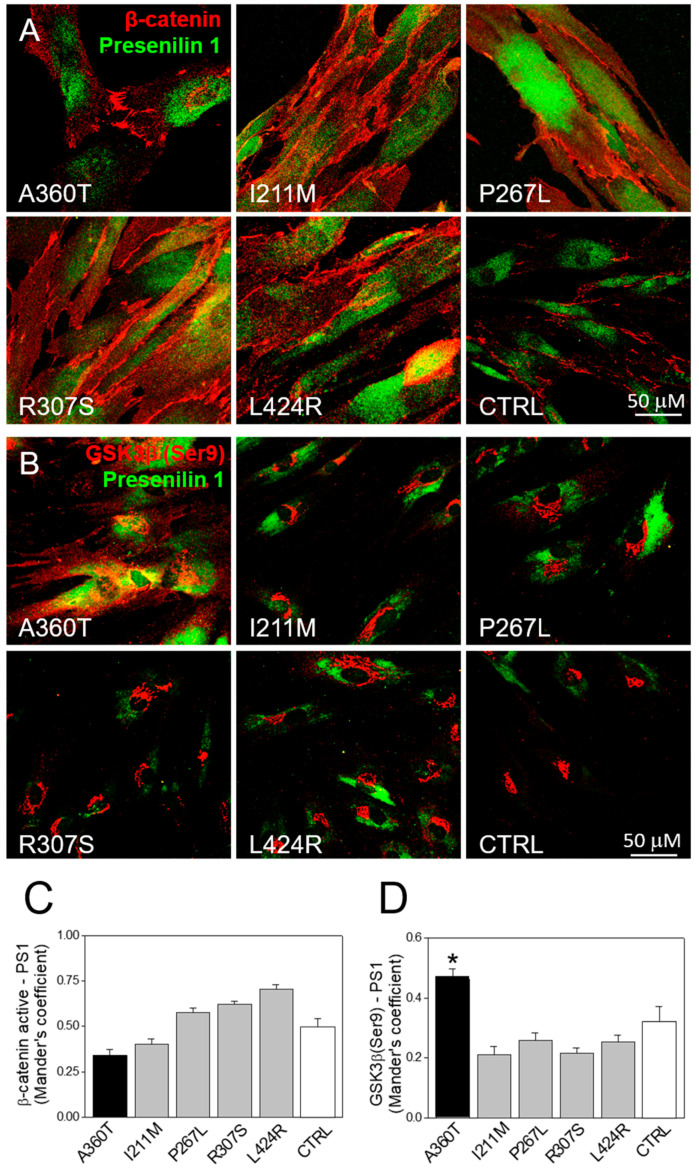
Subcellular localization of β-catenin, GSK3β, and PS1 in A360T fibroblasts. PS1 was co-immunostained with β-catenin (**A**) and GSK3β(Ser9) (**B**) in fibroblasts from A360T patients, other fEOAD patients, and controls. The colocalization extent measured by Mander’s coefficient was weaker for PS1 and β-catenin (**C**) but stronger for PS1 and GSK3β (**D**) in A360T cells (*n* = 3, * *p* < 0.05).

**Figure 15 ijms-24-16999-f015:**
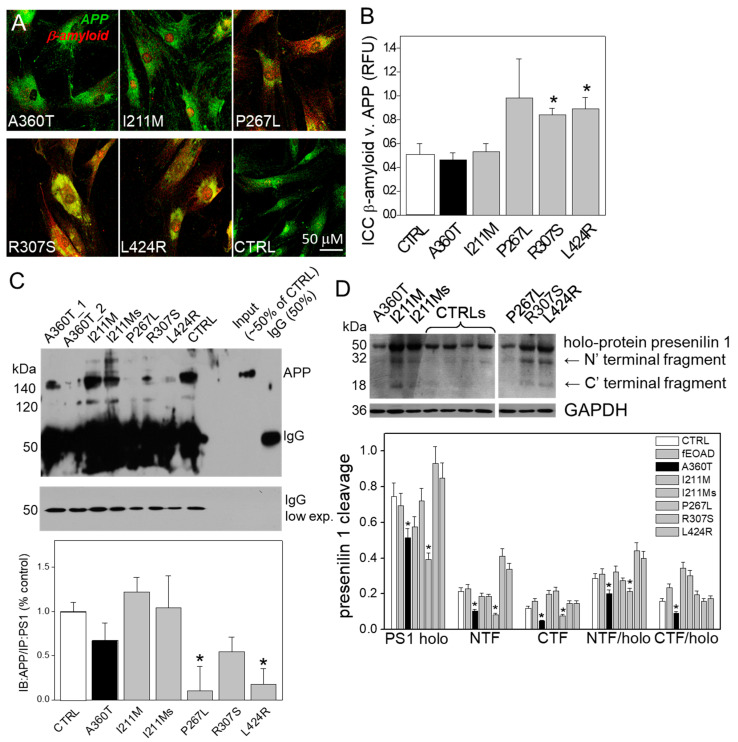
β-amyloid in A360T fibroblasts. Immunostaining indicated a lower content of β-amyloid (in red) in A360T cells than in other fEOADs and control fibroblasts (**A**), expressed in relative fluorescence units (RFU) (**B**). This was consistent with the content of APP in complex with PS1, which was low in R307S, L424R, and P267L cells and markedly higher in the other cells, including A360T cells, *n* = 3, * *p* < 0.05 (**C**). This was accompanied by a lower content of full-length PS1 and its poorer processing into NTF and CTF in A360T than in other fEOADs and control fibroblasts, *n* = 3, * *p* < 0.05 (**D**).

**Figure 16 ijms-24-16999-f016:**
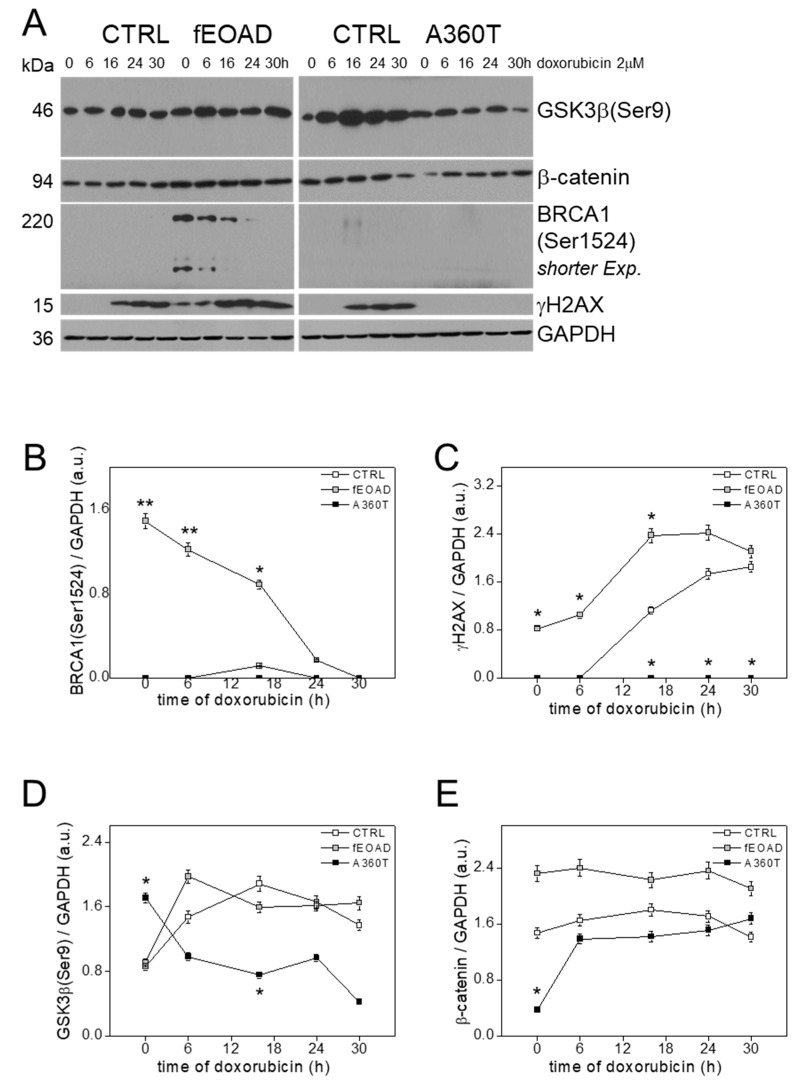
DNA damage response in A360T cells. Fibroblasts were treated for 0, 6, 16, 24, or 30 h with 2 µM doxorubicin (DOXO) and immunoblotted for BRCA1(Ser1524), γH2AX, β-catenin, and GSK3β(Ser9) (**A**). All signals for BRCA1(Ser1524), γH2AX, β-catenin, and GSK3β(Ser9) were quantified densitometrically and standardized to the GAPDH level. Data represent mean values ± SEM, *n* = 3, * *p* < 0.05, ** *p* < 0.01 (**B**–**D**).

**Figure 17 ijms-24-16999-f017:**
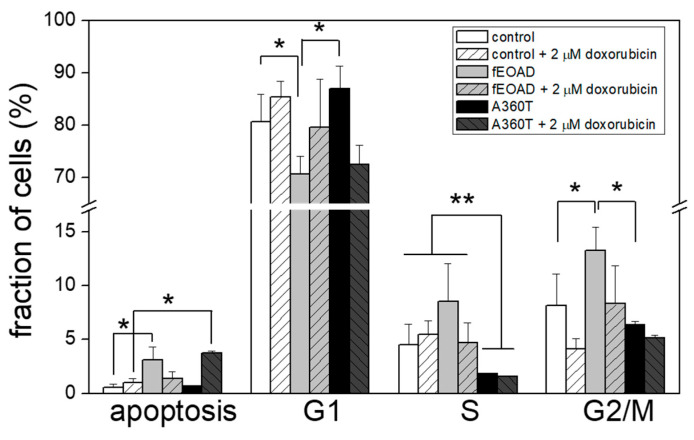
Apoptosis and cell cycle progression in A360T cells under normal conditions and upon DNA damage response. Fibroblasts were treated with doxorubicin for 0 or 30 h, stained with iodide propidium, and subjected to flow cytometry on a CELLQuest Becton Dickinson system. Data represent mean values ±SEM, *n* = 3, * *p* < 0.05, ** *p* < 0.01.

## Data Availability

RNA sequencing data are available at the Sequence Read Archive NCBI NIH database as BioProject https://www.ncbi.nlm.nih.gov/bioproject/382346 (accessed on 23 October 2019) and BioSample: 6704895. Other data are available upon request.

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
