# Peer review of "Rare A360T Mutation Alters GSK3β(Ser9) Binding in the Cytosolic Loop of Presenilin 1, Influencing β-Catenin Nuclear Localization and Pro-Death Gene Expression in Alzheimer’s Disease Case"

_ijms, 2023, doi:10.3390/ijms242316999_

Round 1

Reviewer 1 Report

Comments and Suggestions for Authors

Mutations in the PSEN1 gene, encoding presenilin-1 (PS1), are the most common cause of familial Alzheimer's disease (FAD). The A360T substitution, is located close to GSK3β-tar- 23 geted serine residues which was initially reported in the French population. In the current manuscript, Wężyk et al., demonstrated the pathways and processes involved in the A360T case, 41 highlighting the greatest importance of altered Wnt signaling. It is an interesting manuscript. However, there are some comments need to be further considered:

1. It would be perfect to additionally provide patients with clinical brain imaging such as MRI/PET and compare it to the previous cases. 

2. There is no information about this mutant, whether it is rare or common in the other population, please provide and discuss it extensive

3. Since WES was performed on the patient, please list all potential gene mutants regards the patient's clinical. Also, gene/protein network analysis need tp be applied to further support their findings in term mechanism

4. Family tree also needs, any other members tested for the gene?

5. Fig. 1A needs to be reformated to show protein change

6. Fig. 2C, Fig. 3-12 really need to provide each better solution

7. Protein modeling of the mutation should be provided to support their hypothesis before an in vivo appealing

Author Response

Dear Reviewer,

I am writing to express my sincere gratitude for taking the time to review my manuscript titled " The mutant A360T-PS1 retains inhibited GSK3β(Ser9), followed by the nuclear location of β-catenin and induction of pro-death genes in Alzheimer's disease" Your thoughtful insights and constructive comments have been very valuable in shaping the final version, and I am truly appreciative of the effort you put into providing such detailed feedback.

I want to assure you that I have carefully considered each of your comments and have made revisions accordingly, please fid them in the attachment. Your feedback has not only improved the content but has also helped me gain new perspectives on certain aspects of my work. I am confident that the final version is more robust and well-rounded as a result of your contributions.

Once again, thank you for your time, dedication, and thoughtful feedback.

The MRI scans would be indeed valuable in our manuscript. We have asked for such additional imaging. The Collaborating Medical Center in Åšcinawa in Poland has positively considered our request. The expected delivery of the MRI scans complemented with appropriate clinical descriptions was predicted on 10th November 2023. However due to technical problems, there was a delay with the delivery, and it will be delivered by the upcoming week, up to 17th November 2023. Once delivered the appropriate figure will be incorporated in the body of the manuscript, i.e. as Fig. 1C
Thank you for your understanding.

Warm regards,

Michalina Wężyk, PhD

Reviewer 2 Report

Comments and Suggestions for Authors

The paper is well written it concerns AD no major spelling/grammar errors can be accepted as is 

1. specific improvements should the authors consider regarding the methodology: mutations detected by Sanger sequencing cell culture and whole exome sequencing performed as well + immunoblotting  RNA isolation and protein extracts 

2. pls add a reference to J Hardy et al pmid 27025652

3. Please enlarge Figure 3 for better visibility + Fig 12

Author Response

Dear Reviewer,

I am writing to express my sincere gratitude for taking the time to review my manuscript titled " The mutant A360T-PS1 retains inhibited GSK3β(Ser9), followed by the nuclear location of β-catenin and induction of pro-death genes in Alzheimer's disease" Your thoughtful insights and constructive comments have been very valuable in shaping the final version, and I am truly appreciative of the effort you put into providing such detailed feedback.

I want to assure you that I have carefully considered each of your comments and have made revisions accordingly, please fid them in the attachment. Your feedback has not only improved the content but has also helped me gain new perspectives on certain aspects of my work. I am confident that the final version is more robust and well-rounded as a result of your contributions.

Once again, thank you for your time, dedication, and thoughtful feedback.

Warm regards,

Michalina Wężyk, PhD

Round 2

Reviewer 1 Report

Comments and Suggestions for Authors

Thank you the authors for clearly addressing all my concerns, the revised is now extensively implemented. Congratulations!

Author Response

thank you.